# A High Compressibility Pressure—Sensitive Structure Based on CB@PU Yarn Network

**DOI:** 10.3390/s18124141

**Published:** 2018-11-26

**Authors:** Xingtong Chen, Chunguo Liu, Shuo Liu, Bing Lyu, Donglai Li

**Affiliations:** Roll-forging Research Institute, College of Materials Science and Engineering, Jilin University, Changchun 130025, China; chenxt16@mails.jlu.edu.cn (X.C.); liushuo940313@163.com (S.L.); lvbing1613@163.com (B.L.); 15526899829@163.com (D.L.)

**Keywords:** pressure-sensitive, net-like structure, complete compression, responsive behaviors, piezoresistive sensors

## Abstract

In this work, a piezoresistive sensor structure based on carbon black (CB)@polyurethane (PU) yarn material was developed. Specifically, CB@PU yarn was constructed by the polymer-mediated water-based electrostatic deposition method. The distribution of the yarn was artificially controlled to fabricate conductive networks. The CB conductive layer was efficiently supported by the net-like structure of PU yarn, thus generating collaborative advantage. The as-fabricated pressure sensor not only displayed compressibility of over 97%, but also detected a wide pressure change from 25 Pa to 20 kPa. Furthermore, this sensor exhibited response time of less than 70 ms and reproducibility of over 10,000 cycles. The advantages of the CB@PU network ensured this pressure-sensitive structure enormous potential application in pressure sensitive equipment.

## 1. Introduction

Pressure sensors with wide detection range and high compressibility are highly desirable in future portable devices. Recently, various pressure-sensing mechanisms including transistor sensing [1,2], capacitive sensing [3,4], piezoelectric sensing [5,6], triboelectric sensing [7,8,9] and resistive sensing [10] have been used to construct pressure sensors with excellent performance. Resistive-sensors in particular draw extensive attention due to their simple device structures and low energy consumption.

Resistive-sensors’ functions are based on bulk piezoresistivity and the change in contact resistance (R_c_). Limited by their unstable mechanical structures and compression volume, current piezoresistive sensors cannot meet the needs of some emerging applications which require significant elastic compression [11]. Shi and co-workers [12] fabricated sensors with microcracks based on graphene-covered human hairs. However, the finite value of electrical conductance suddenly dropped to zero when the microcracks were disconnected. Such sensors based on the pure crack mechanism exhibited limited detection range because of the fragile conductive pathways. Wu and colleagues prepared strain sensors based on the compressible structure of CB@PU sponges. However, when the applied strain to sponge reached a limiting value (60%), the sensors could not be compressed more or detect further [13].

Conventional conductive polymer films are potential materials for resistive-type sensors, which can detect large compressive strains. However, these materials are insensitive, unstable, and incapable of detecting low pressures [14,15,16]. In order to improve the performances of resistive-sensors, microstructure designs were used, such as fractured microstructures [13,17], reversible interlocked microstructures [18], micro-pyramid array structures [19] and carbon nanotube mesh structures [20]. These devices could precisely detect low pressure stress, but most of them could not be applied to relatively high pressure regime (>10 kPa). For example, a previously reported pressure sensor using an Au-coated micropillars array structure showed a high sensitivity of 2.0 kPa^−1^ in the pressure range below 0.35 kPa, but it could not detect pressures above 3.5 kPa due to the current saturation [21]. Another work based on a microcracks structure showed a minimum detectable pressure of 9 Pa, but the effective range of the sensor was less than 2 kPa [16].

Therefore, it is very challenging to fabricate a pressure sensor with good compressibility which is applicable to a broad detection range. The network structure presents advantages such as structural stability, excellent integrity and good interaction within the network. When the network is given elasticity, small external pressure variation can cause significant elastic fluctuation, and the elastic network can produce large deformation under strain. Meanwhile, polyurethane (PU) materials are widely utilized for pressure sensors due to their elastic elongation, good mechanical property and chemical stability [15,16,22,23,24]. In previous studies, polyurethane pressure sensors were mainly manufactured in forms of sponge or foam, but polyurethane yarn have rarely been reported in pressure sensors by far. PU yarn’s excellent elasticity makes it a desirable material for pressure sensors. Thus, it is worthy of further exploration and utilization.

In this paper, a new pressure sensor structure was proposed. Combining the advantages of PU fibers and net-like structures, a highly elastic and sensitive 3D network sensor was fabricated. Through the electrostatic deposition method, the CB conductive layer was efficiently supported by the PU yarn. The detection range of pressure was significantly extended (from 25 Pa to 20 kPa) and the tolerable compressive strain was as high as 97.4%. Furthermore, high sensitivity (2.3 kPa^−1^), fast response time (70 ms) and good reproducibility over 10,000 cycles endowed this material with wide potential applications in pressure sensitive devices.

The rest of this paper was organized as follows: material preparation and fabrication methods for yarn network are described in Section 2. In Section 3.1, the structural features of the CB@PU yarn network are described. The conduction mechanisms of the CB@PU network are addressed in Section 3.2, followed in Section 3.3 by a presentation of the piezoresistive characteristics of CB@PU network pressure sensors, and finally, our conclusions in Section 4.

## 2. Materials and Methods

### 2.1. Materials

Nano conductive carbon black (BP 2000) was provided by Cabot Chem (Shanghai, China). The NR latex was provided by Chengdu Xinyuanding Co., Ltd. (Chengdu China). Polyurethane yarn was purchased from Dongguan Sheng Sheng Plastic Chemical Co., Ltd. (Dongguan, China). Cellulose nanowhisker (CNC) was purchased from Shanghai Sciencek Co., Ltd. (Shanghai, China). Chitosan (CS) was purchased from Chengdu Kelong Chemical Reagent Company (Chengdu, China).

### 2.2. CB@PU Conductive Network Manufacturing Process

The elastic networks were fabricated by interweaving polyester fibers (12 tex, 34 tex, 73 tex) evenly in a porous ring (d = 32 mm, h = 11.5 mm, v = 9.25 cm^3^). The arrangement of holes on the ring was four rows by 24 columns. The holes in the same row or column were equally spaced respectively. The ring was longitudinally divided into four segments, and each segment was between the two rows of holes. PU yarn was wound segment by segment. In the longitudinal direction, the yarn was repeatedly interweaved between the holes in opposite columns in an “X” shape. In the transverse direction, the yarn was arranged in a “Z” shape. Each layer of PU yarn was 0.9 m. The total length of PU yarn was 3.6 m. In order to remove surface impurities, the PU yarn was cleaned with ethanol. For preparation of CB@CNC suspension [25], exactly 0.2448 g CB was added to 800 mL CNC (0.613 mg·mL^−1^) suspension, and the mixture was sonicated for 30 min to yield a fully suspended negatively charged CB@CNC suspension. Then 3.8 g NR latex (NR content was 35%) was added into the CB@CN suspension (the mass ratio of CB/CNC/NR was 1/2/15.5), and the CB@CNC/NR mixture was mixed uniformly. CS solution (0.8 mg·mL^−1^) was prepared as a charge mediator. Positively charged stabilized CB@CS (the mass ratio of CB/CS was 1/2.6) suspension was prepared in the same way as mentioned above. The cleaned PU yarn network was immerged into CB^+^ suspension, forming negatively charged CB@PU network. Then the negatively charged PU yarn network was immerged into CB^+^ suspension. This immersion cycle was repeated different times to achieve the deposition of CB. In this process, the CB@PU elastic network was dried after every ten cycles. Finally, the obtained CB@PU network was dried at 80 °C for 6 hours to remove moisture.

### 2.3. Characterization

The sensor resistance and yarn resistance was measured using a 124 oscilloscope (Fluke, Washington, DC, United States). Scanning electron microscopy (SEM) was performed with a JSM-5900LV microscope (JEOL, Tokyo, Japan). The current signal of the pressure sensor was measured with a source meter (mod. 2420, Keithley, Cleveland, OH, United States). The compression property was measured with a digital thickness gauge. The mechanical properties and deformation of the pressure sensor were tested under a multi-functional tester (Instron-5560, Boston, MA, United States).

## 3. Results and Discussion

### 3.1. Structure and Preparation of CB@PU 3D Network Structure

Figure 1a illustrates the fabrication process of the CB@PU yarn network. Figure 1b,c show the spatial structure of the sensor. The holes on the ring were evenly arranged. PU yarn was wound segment by segment in order to ensure equal length in the unit space, and the yarn was wound according to certain rule. A cross section of Z-shaped arrangement yarn is shown in Figure 1d,e, which reveals the longitudinal interweaved layout of the yarn. In a word, the uniform distribution of the network structure could be controlled. Positively charged chitosan (CS) and negatively charged cellulose nanocrystals (CNC) were used as mediators. The PU elastic network was immersed into a positively charged CB@CS solution and a negatively charged CB@CNC solution successively. During electrostatic deposition, CB particles were gradually coated on the surface of the PU yarn network. After drying, the final CB@PU conductive network was fabricated.

Compared with the original white PU yarn network (Figure 2a), the CB@PU network was darker after loading (Figure 2b). The dark visual appearance indicated that CB achieved a good load. SEM was used to observe the surface characteristics of PU yarn and of CB@PU yarn. As addressed in Figure 2c,d, PU yarn (34 tex) had a diameter of 80 μm and consisted of a bunch of PU microfibers. The surface of PU yarn was very clean and smooth before deposition of CB. In Figure 2e,f, it can be seen that the surface of the PU yarn became rugged after CB deposition. It can be clearly seen that the CB nanoparticles were well distributed. These indicated that the CB layer can be coated on the PU yarn via an electrostatic deposition method.

The CB@PU network was tested for compression. Figure 3a,b show images of the experimental sample before and after compression. The resulting CB@PU fiber network exhibited excellent flexibility in the compression process and achieved a compression deformation of 97.4%. This was attributed to the combination of the elastic PU yarn material and the hollow structure of the 3D yarn network.

PU yarn possesses excellent flexibility and elastic elongation. Compared to structures of other sensors fabricated or assembled with entity materials, the hollow yarn mesh skeleton is not limited by the volume of the material, which can be almost completely compressed. Such properties have great advantages in testing large strains.

### 3.2. Pressure-Sensing Mechanism

When increasing pressure is applied upon the CB@PU yarn network, the bending of the conductive PU yarn will cause the deposited CB conductive layers to deform and mutually contact, leading to a change in electrical resistance and the electrical signal. Most piezoresistive sensors, such as polymer sensors and flexible sensors, deform as a whole under pressure. But the CB@PU network sensor is not directionally compressed. That is, the sensor can be divided into two parts: compressed part and uncompressed part. During the press-down process, the upper layer is subjected to a direct force and is compressed. The yarns in the upper layer closely contact with each other and then form a dense compressed layer. The uncompressed part below will not be affected by the external force. The yarns in this part are still separated from each other, so the density of the yarns in network will not change. This isillustrated in Figure 4.

For the compressed layer, there are two types of deformation in the conductive network structure: deformation of the macro PU network and of the microscopic CB conductive layer. From the microscopic point of view, when the PU network is compressed, the bending of yarn causes tension on the CB layer. The CB nanoparticles’ distribution on the conductive layer becomes sparse, and the electrical resistance of conductive yarn will increase. After releasing compression strain, the PU yarn can still retain its original length. The CB layer merges and returns to a tight particle distribution, which causes the decrease of resistance. This reversible stretching deformation of CB conductive layer generates a corresponding change in yarn’s conductivity. It can be calculated that the maximum elongation of a single yarn in the network is 20.6%. The specific calculation process is explained in Figure 5 and the following equations.

When the yarn is compressed from the uppermost layer to the lowermost layer, the strain of the yarn is greatest. The height of the PU yarn network was 11.2 mm before compression. The height of the PU yarn network was 0.29 mm after a full compression, so the maximum height difference was 10.9 mm:$$\;\sqrt{16^{2}+10.9^{2}}=19.3\;$$
(19.3 − 16)/16 × 100% = 20.6%
therefore the maximum length change of the single yarn in network was 20.6% after a full compression.

The Fluke 124 oscilloscope was connected with the both ends of yarn to measure the resistances of single CB@PU yarn at different strains. Figure 6 reveals that the relative resistance of a single CB@PU yarn corresponded to a gradual increase of the strain. In the process of making network structure, each yarn in the network was stretched to 200% of the original length it was, and when the sensor was unstressed, the yarn in the network already had a 100% strain, so when the sensor was under pressure, the calculations of amount of deformation and of the corresponding change in resistance were both based on the 100% strain which had been generated. The corresponding resistance change rate could be obtained through the following equation:200%×(1+20.6%)=241.2%, i.e., 141.2% strain
So the corresponding resistance change rate was: (6.75−5.34)/5.34×100%=26.4% 
The data is displayed in Figure 6.

However, the sensing mechanism is mainly based on the macro change in the network. Macroscopically, due to the high density of the yarns in network (up to 40.71 cm per cubic centimeter, which could be calculated based on the sensor volume and total length of yarns), just under a slight pressure, the yarns would mutually contact and generate dense point contacts and line contacts on CB layers. Yarns in the compressed layer overlap with each other, CB particles contact with each other and form chain-shaped conductive channels which can be seen in Figure 7a. Electrons could move along these channels to form a current. Since such contacts are so dense that a plurality of conductive channels act simultaneously, the carbon black forms a spatial network chain structure in the compressed layer. The phenomenon is regarded as percolation transition [26,27]. As shown in Figure 7d, the current flows straightly from top to bottom of the compressed layer through spatial network chain structure instead of flowing along the conductive yarn (Figure 7c), and the resistance drops sharply. The percolation theory is the main conductive mechanism of the compressed layer and will be demonstrated later. There is also a quantum tunneling effect [28,29,30,31,32] in the compressed layer. As the pressure increases, the distance between carbon black particles decreases gradually. When particles are close enough but not in contact, due to the volatility of microscopic particles, some electrons will penetrate the energy barrier between adjacent carbon blacks, and then there will be a quantum tunneling effect. Figure 7b illustrated this effect.

In order to verify the tunnel mechanism, we tested current-voltage behavior of samples with immersion cycles of 10 times and 20 times at 5.2 kPa and 20 kPa, respectively. Theoretically, there must be a quantum tunneling effect between the carbon black particles in the network structure and tunneling mechanism could lead to non-ohmic behavior. However, linear I–V relationships were observed which didn’t match the theory. For this mismatch, we suspect that it was because even though the number of immersion cycles was small, the concentration of carbon black was still high, so it was likely that the effect of quantum tunneling was obscured by the percolation effect. In subsequent experiments, we will improve the experimental setup and explore this further.

In order to prove the main theoretical mechanism of the compressed layer—the percolation theory, the resistance of the 0.29 mm thick compression layer at different immersing cycles was measured, as illustrated in Figure 8.

Obviously, as the number of immersion cycles increased, the resistance exhibited an exponential change. This trend is consistent with the exponential function form of the classical percolation theory formula. According to the classical percolation theory, the volume resistivity of the conductive composite and the volume fraction of the conductive filler satisfy Equation (1).
(1)$$\;\sigma =\sigma_{0}{\left(\varphi -\varphi_{c}\right)}^{-t}\;$$
where σ is the resistivity of the composite, φ represents the volume fraction of the CB in the composite, $\varphi_{c}$ reveals the critical volume fraction (volume fraction at percolation), $\sigma_{0}$ and t are fitted constants. Theoretical predictions of the critical exponent, t, range from 1.6 to 2.0 [33,34] while experimental values between 1.3 and 3.1 have been reported [33].

The relevant experimental data were measured and integrated in order to be matched with the classical percolation theory formula for further demonstration of the percolation effect in the CB@PU network. Firstly, the volume fraction of CB in the CB@PU network at different immersing cycles were calculated. The calculation method was as follows: the solid matter loaded on the yarn network, including CNC, CS, CB, NR, and their mass ratio was equal to their mass ratio in the suspension. CB/CNC/CS/NR = 0.49/0.49/0.64/1.33. The weight gain of the CB@PU network after 100 immersion cycles was 1.15 g. The mass of CB was proportionally calculated to be 0.191 g. It was considered that the CB of each load was of the same mass, so the CB content per cycle was 0.00191 g. The BP2000 carbon black had a density of 144 g/L, i.e., the CB volume per load was 0.0132 mL. The CB volume fraction at different immersion cycles thus could be calculated.

According to Equation (2), the relationship between the electrical resistivity of compressive layer with a thickness of 0.29 mm and the carbon black content was obtained, as shown in Figure 9.
(2)$$\;R=\sigma \frac{l}{S}\;$$
where R is the sensor resistance, σ represents the sensor resistivity, l represents the height of sensor, S reveals cross-sectional area of sensor.

The solid line in Figure 9 is the curve of the resistivity of the compressed layer as a function of carbon black content according to percolation theory (Equation (1)). The relevant parameters of the fitting were as follows: σ_0_ = 936,375 Ω·cm, φ_c_ = 6.626%, t = 2.8, R^2^ = 0.990, R^2^ is the fitting correlation coefficient. The critical exponent of the compressed layer was in reasonable agreement with both the experimental and theoretical predictions. It could be seen that the experimental data shows a good fit with Equation (1), which provides a basis for the mechanism of this sensor.

It is worth noting that the current flows from top to bottom through the compressed layer vertically instead of flowing along the yarn, which means that the change of resistance of the yarn can be reflected only at the intersections of overlapping yarns. What’s more, the maximum length change of the single yarn in network was 20.6% and the corresponding resistance change rate was less than 27%. Therefore, in the compression process, the change of single yarn resistance had an effect just at intersections and would have little impact on the whole network.

For the uncompressed layer, the yarns are not subjected to external force directly and will not deform. The distance between the CB conductive layers is too large to form a conductive path or quantum tunneling. The current is mainly conducting along the conductive yarn, as revealed in Figure 7c, so the uncompressed layer has a high electrical resistance. Since the yarns in the network are evenly distributed, the length of yarn per unit space is approximately equal, so the resistance value of the uncompressed layer is determined by its height. During the compression, the uncompressed layer of the sensor gradually transforms into the compressed layer and the total resistance of the sensor decreases sharply during the press-down process.

The sensor (with 100 immersion cycles) was subjected to different pressures and produced different strains. As illustrated in Figure 10, the upper and lower sides of CB@PU network sample were covered with 24 × 24 × 0.1 mm copper sheets. The copper wires were bonded with silver paste in order to remove contact resistance and obtain a stable output signal. A small copper slice was inserted at the junction of uncompressed layer and compressed layer to test the resistance of the two parts, respectively. Table 1 shows the resistance values of the two parts under different compressions. It could be seen that compressed layer resistance was much smaller than uncompressed layer resistance which approximated the resistance of the entire sensor. Therefore, the entire sensor could be regarded as a uniform resistance, ignoring the compressed layer resistance. Its resistance was only determined by the height of the uncompressed layer.

To understand this resistivity behavior, a model was constructed to describe it. The resistance value of the sensor with different deformations were calculated through the resistance value formula. R_p_ is the resistance of the sensor in the compression process and R_p_ satisfied Equation (3):(3)$$\;R_{p}=\;\rho \frac{x_{0}-x}{S}+R_{c}\;$$
where ρ is the resistivity of the sensor, x_0_ represents the height of the sensor, x indicates the deformation and S is the cross-sectional area of the sensor. R_c_ is the sensor resistance correction coefficient used for correcting errors caused by yarn uniformity, vibration during compression, and the resistance of compressed layer.

Due to the homogeneity of the yarn network, r remained constant during compression. Calculus idea was used to derive the relationships between the resistance value or the resistance change rate φ and the amount of compression. As shown in Equation (4) and Equation (5), respectively:(4)$$\;{\;R}_{p}=\int_{x}^{x_{0}}\mathrm{rdx}+{\;R}_{c}=r\left(x_{0}-x\right)+{\;R}_{c}\;$$
(5)$$\;\emptyset =\frac{R_{P}-R_{0}}{R_{0}}+\;\frac{R_{c}}{R_{0}}=-\frac{\mathrm{rx}}{{\mathrm{rx}}_{0}}+\;\emptyset_{c}={\mathrm{GF}}_{0}\cdot x+\;\emptyset_{c}\;$$
where GF_0_ is the theoretical gauge factor of the sensor, GF_0_ = $-\frac{1}{x_{0}}$. $\emptyset_{c}$ represents the correction factor of change rate of the resistance.

### 3.3. Piezoresistive Characteristics of CB@PU Network Pressure Sensors

At a constant voltage, the electrical signal of the CB@PU elastic network was recorded as a function of deformation. The theoretical relationship between resistance change rate and deformation was expressed as Equation (5), $\emptyset ={\mathrm{GF}}_{0}\cdot x+\;\emptyset_{c}$, GF_0_ = $-\frac{1}{x_{0}}$. The resistance decreased after pressure applied to the surface of the copper sheet. The measured value was calculated and the actual function of deformation vs. resistance change rate was plotted.

Figure 11a–c show the sensitivity of CB@PU network sensors fabricated with different PU yarns (12 tex, 34 tex, 73 tex with 100 immersion cycles) to the applied strain. The data are fitted according to Equation (5) to get the corresponding curve. The corresponding coefficients are listed in Table 2. The data shows that the change rate of resistance varies linearly with the amount of compression. The gauge factor (GF), which is usually defined as the ratio of the relative change (ΔR/R_0_) and strain, was measured. The linearly dependent coefficients R_2_ of the three samples are all greater than 0.99, so GF and GF_0_ can be considered as approximately equal. It could be seen that the experimental data were highly fitting with Equation (5). These data revealed that the sensitivity of compressive strain was constant throughout the strain process, which verified the correctness of the sensing mechanism. Furthermore, the stable structure of the sensor enabled to accurate prediction of the deformation via changing sensor resistivity. In addition, since the distribution of PU yarns in the network structure was controllable, the distribution and winding method of yarns could be designed according to different requirements. This structural characteristic will be further explored in future research.

Figure 12 shows the sensitivity of CB@PU network sensors fabricated with different PU yarns (12 tex,34 tex,73 tex with 100 immersion cycles) to the applied pressure. The sensitivity of the pressure sensor was defined as $S=\left(R_{0}-R_{P}/R_{P}\right)/\Delta P$, where ΔP is the change of applied pressure, and R_0_ and R_P_ represent the sensor resistance after no pressure and applied pressure, respectively. 

The points in Figure 12 were divided into four regions according to the sensitivity difference. They were the micro-pressure region (purple region, 0–140 Pa), low- pressure region (yellow region, 140–1.07 kPa), the medium-pressure region (pink region, 1.07–17.1 kPa) and the high-pressure region (blue region, 17.1–20 kPa). It could be seen that from the low-pressure to the high-pressure region, the sensitivity of the sensors decreased gradually. This trend was attributed to two factors. Firstly, when the PU network deformed, the elastic potential energy accumulated and required more pressure for further deformation. Secondly, the increase of the strain caused the resistance rate to decrease linearly. The combination of the two factors caused the sensitivity to decrease gradually from the micro-pressure zone to the high-pressure zone.

As the pressure increased, the resistivity curves showed a similar trend. However, the thinner PU yarn in the micro-pressure zone showed higher sensitivity of 2.33 kPa^−1^. This was because the yarns in different sensors had the same density per unit area. The thicker the yarn, the larger the area superimposed between the yarns, and the larger the effective contact area of the conductive layer. From no-pressure to micro-pressure, the network with thicker yarn had a more obvious current change. In the remaining regions, the sensor made of finer PU yarn had higher sensitivity. The yarn with thinner network had smaller elastic modulus and softer structure that could be easily compressed, so the sensitivity was higher. What’s more, the experimental results showed that the thinner yarn had higher GF but lower linearity. The resistivity curve of the 73 tex yarn was more stable than that of the 12 tex yarn. The stiffness of the thick yarn network was higher and it could afford higher force, thus the stability of the network was higher.

The CB@PU network fabricated with 12 tex PU yarn was preferred. The resistance response upon cyclic loading was studied to evaluate the potential of the CB@PU network as a strain sensor. Figure 13 illustrates the hysteresis curve for the CB@PU network sensor. 

As revealed in Figure 13, for small stress such as σ = 500 Pa, the relative change of the resistance overlapped with that in the stress-release cycle for the sample, indicating a small hysteresis in the response. For a larger stress such as σ = 10 kPa, there existed a relatively large hysteresis in the response. However, even in this case, the original resistance of the sensor was fully recovered after releasing it from strain.

It should be mentioned that in the process of making network structure, each yarn in the network was stretched to 200% of the original length, that was, when the sensor was unstressed, the yarn in the network had a 100% strain, so the yarn had an elastic potential energy in the rest state. Therefore, when subjected to a pressure, the reaction time of the sensor and the rebound time of the yarn were thus reduced. The hysteresis properties caused by PU yarn materials were alleviated. In the subsequent research process, we will continue to explore methods to improve the hysteresis.

The elastic performance was tested and the capability of the device as a strain sensor was evaluated. The responsive behaviors of CB@PU networks in different regions were recorded. The normalized current response of repeated pressure loading and unloading cycles was recorded and plotted in Figure 14a,b. As the pressure increased from 25 Pa to 1250 Pa, the responsive current signals became more and more obvious. As seen in Figure 14a, stable and continuous current responses were clearly visible when the detected pressure was as low as 25 Pa. This minimum detectable limit was lower than some recently reported sensors [8,35]. The high sensitivity of pressure sensors to small pressure change could be attributed to the soft structure of the elastic network. Light touching could cause a significant contact between the yarns and produce sensitive responses. These responses of sensors to small pressures indicated their ability to detect small movements.

In the repeated high-pressure region, the current curve of the sensor was obtained and plotted in Figure 14b. The intensity as well as the shape of these signal peaks varied for different pressure values. The higher the pressure, the sharper the peaks would be. The current signal was still relatively stable within the pressure range of up to 20 kPa. This maximum detection limit was higher than most of the known small piezoresistive sensors. Figure 14c shows a fast response time (<70 ms) of the sensor, which was similar to the reported response speed of some other sensors [35,36,37]. The reproducibility of the CB@PU network was also evaluated. As shown in Figure 15a, when a pressure of 5 kPa was applied to the surface, the resistance of the sensor remained stable after 10,000 loading-unloading cyclic tests. Figure 15b shows that the sensor had a high degree of reproducibility which was attributed to the desirable compression-resilience property of the PU network and the firmness of the CB conductive layer load. This meant that the pressure sensor with this structure had a long service life and high degree of stability.

## 4. Conclusions

We have developed a new pressure-sensitive structure based on the CB@PU elastic network. An even 3D network structure was constructed by controlling the distribution and direction of yarn in the network artificially. The combination of the network sensing mechanism and the CB@PU conductive yarn endowed the sensor with high compressibility (97.4%) and capability of wide range detection (25 Pa to 20 kPa). Notably, the CB@PU pressure sensor exhibited excellent flexibility, fast response time (<70 ms), excellent sensitivity (up to 2.3 kPa^−1^) and good reproducibility (over 10,000 cycles). This pressure-sensing platform is comparable to the recently reported equipment in terms of overall performance, but it has significant advantages with regards to measurement range and compression performance.

## Figures and Tables

**Figure 1 sensors-18-04141-f001:**
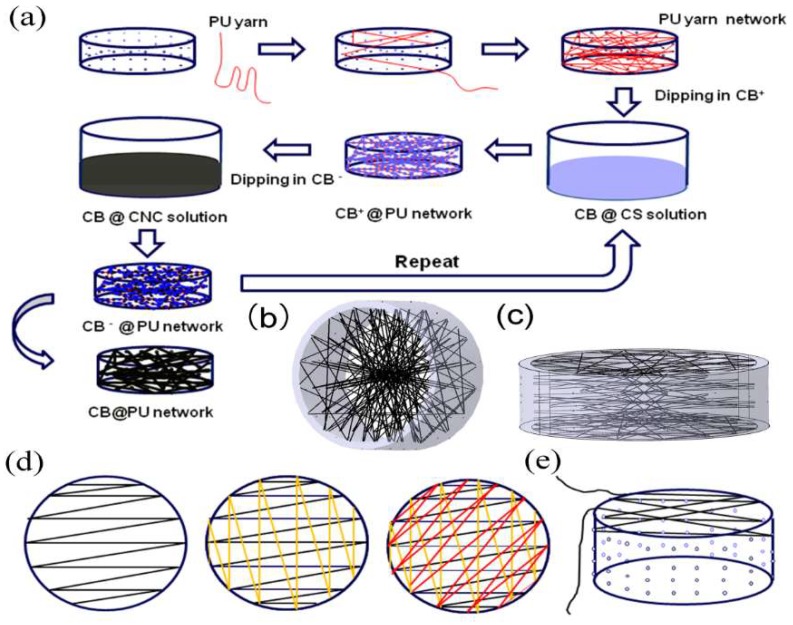
(**a**) Schematic illustration for preparation of the CB@PU yarn network assembly process; (**b**,**c**) Schematic illustrations of the PU yarn network. Schematic diagram of the horizontal interlacing method (**d**) and the longitudinal interlacing method of the yarn network (**e**).

**Figure 2 sensors-18-04141-f002:**
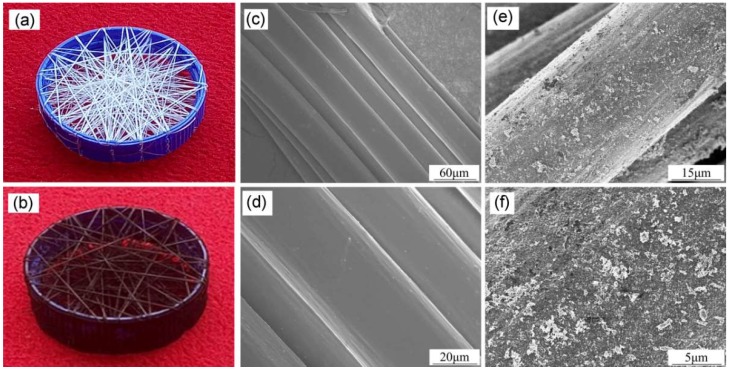
(**a**) Photographs of neat PU yarn network and (**b**) CB@PU yarn network.SEM images of a neat PU yarn (**c**,**d**) and CB@PU yarn (**e**,**f**) under different magnifications.

**Figure 3 sensors-18-04141-f003:**
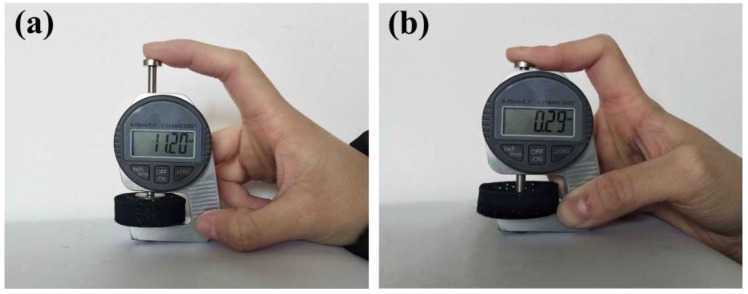
(**a**,**b**) Photographs show the compressibility of CB@PU network.

**Figure 4 sensors-18-04141-f004:**
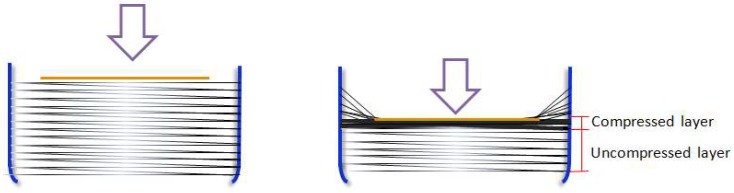
Schematic evolutions of the CB@PU network during continuous compressive deformation.

**Figure 5 sensors-18-04141-f005:**
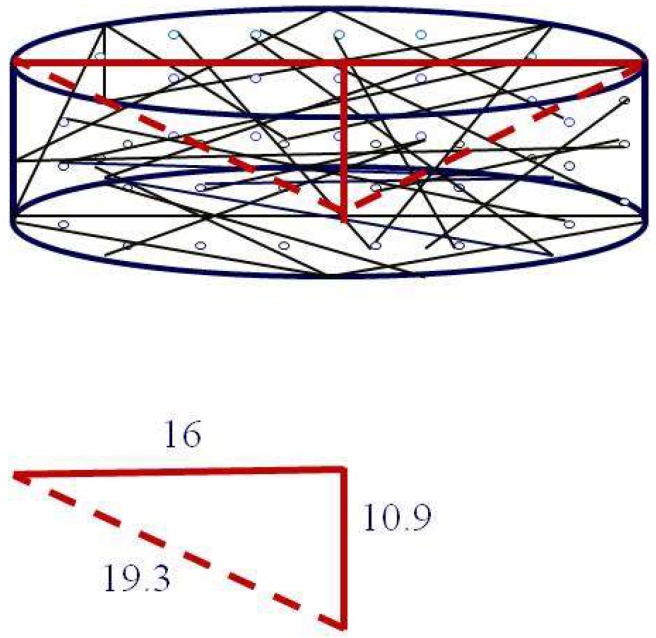
Schematic diagram of the length change of single yarn in CB@PU network under a compressive deformation.

**Figure 6 sensors-18-04141-f006:**
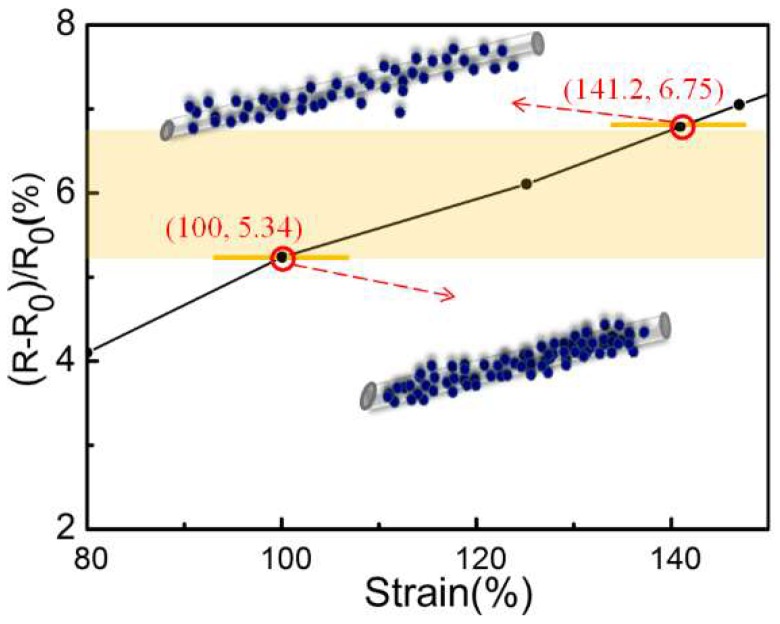
A single CB@PU yarn corresponds to a gradual increase in the relative resistance of the strain.

**Figure 7 sensors-18-04141-f007:**
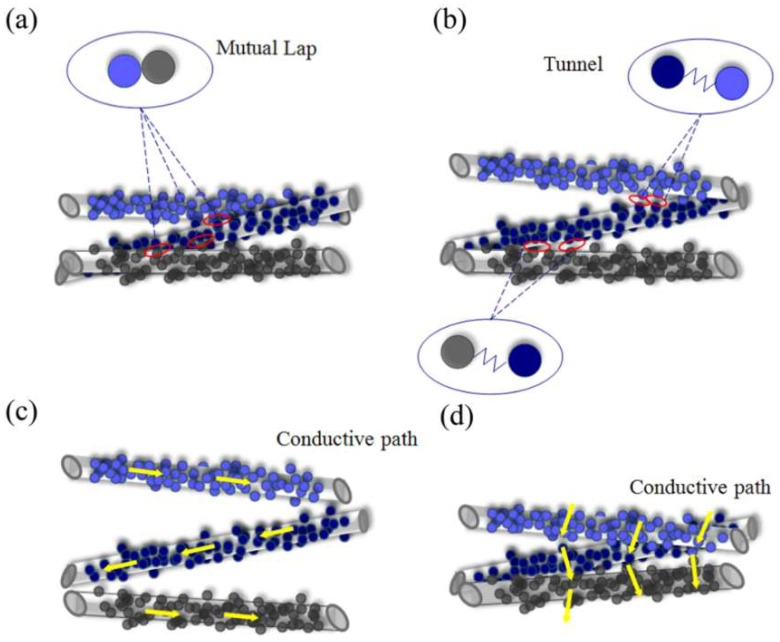
(**a**) Illustration ofpercolation transition and (**b**) quantum tunnel; (**c**) Transfer path of current in uncompressed layer and (**d**) in compressed layer.

**Figure 8 sensors-18-04141-f008:**
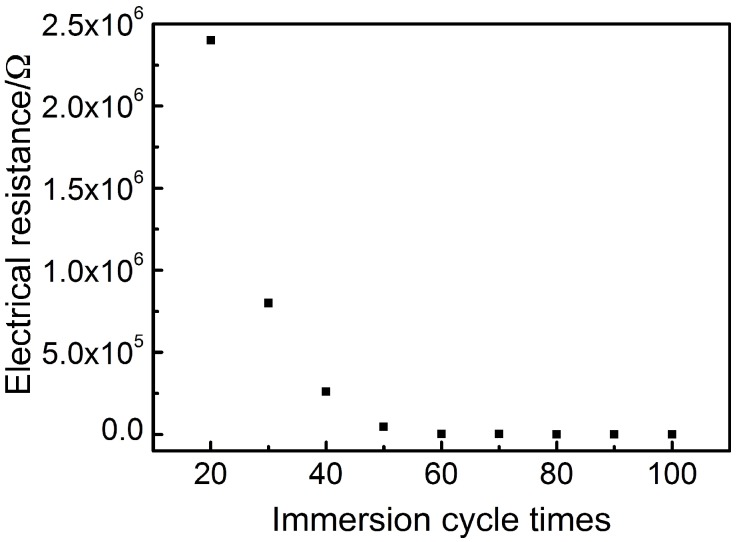
Electrical resistance change of CB@PU yarn with increasing immersing cycles.

**Figure 9 sensors-18-04141-f009:**
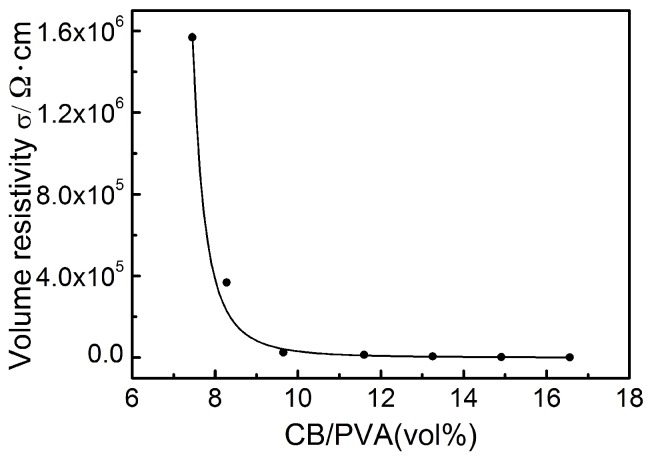
Percolation equation fit to the experimental data (conductivity data from Figure 8).

**Figure 10 sensors-18-04141-f010:**
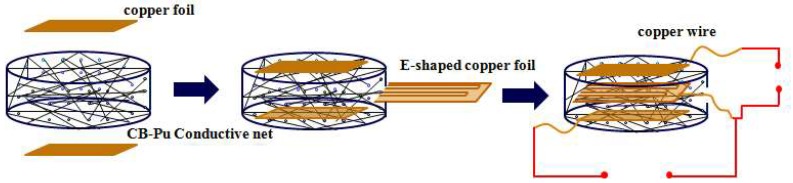
Schematic diagram of the structure of the pressure sensor.

**Figure 11 sensors-18-04141-f011:**
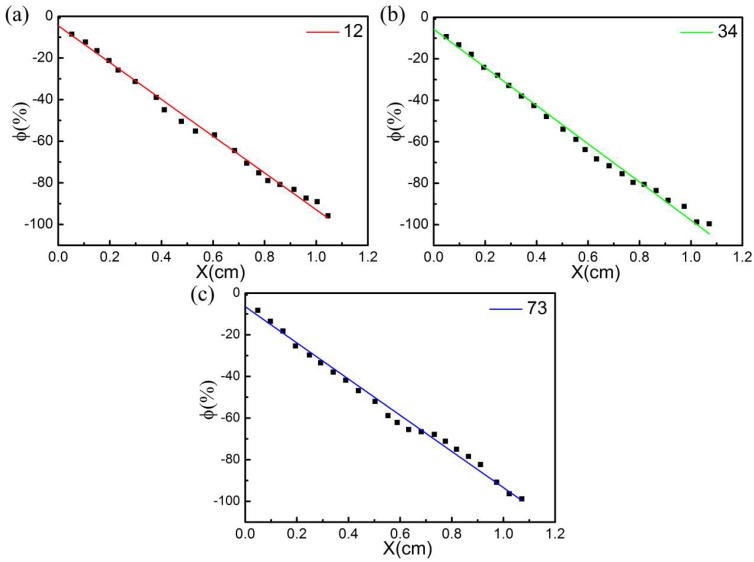
Change of resistance variation rate with strain under PU yarn density of 12 tex, 34 tex and 73 tex (with 100 immersing cycles). Fitting formula of (**a**):∅ = −0.8842x − 4.5025; Fitting formula of (**b**): ∅ = −0.9207x − 5.7600; Fitting formula of (c): ∅ = −0.8689x − 6.5401.

**Figure 12 sensors-18-04141-f012:**
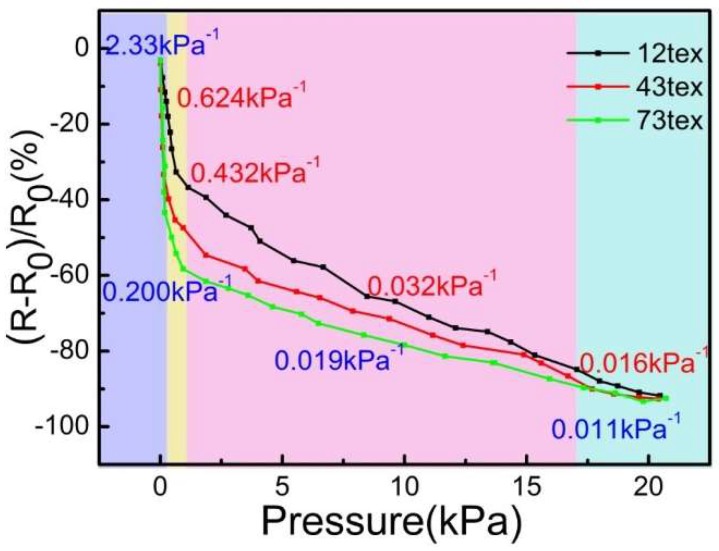
Pressure sensors with different yarn linear density respond to increasing pressures.

**Figure 13 sensors-18-04141-f013:**
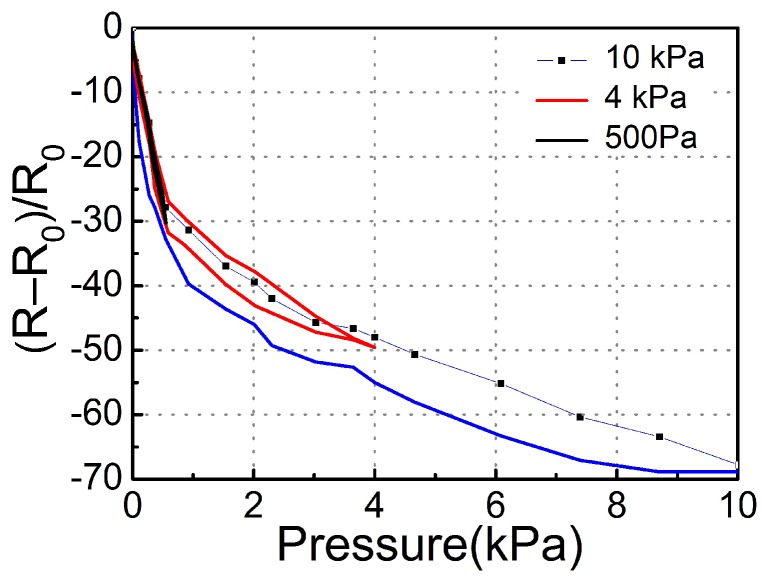
Hysteresis curves for the CB@PU network structured sensor.

**Figure 14 sensors-18-04141-f014:**
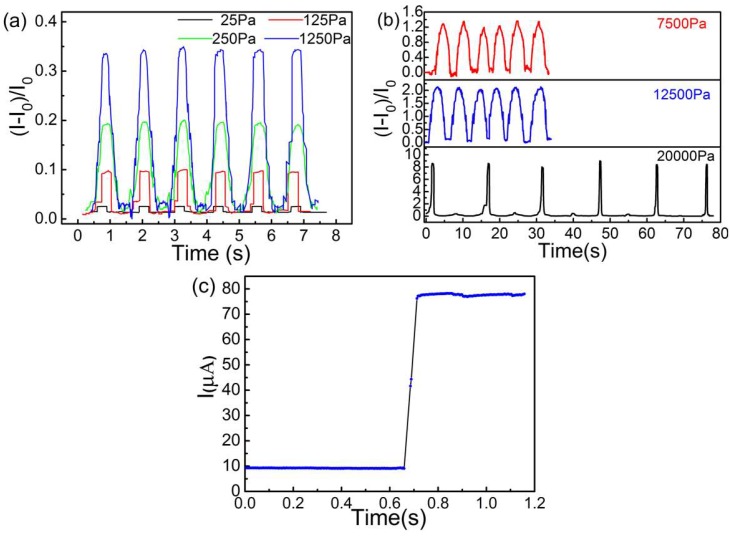
(**a**) A pressure sensor with 12 tex PU yarn corresponds to the relative resistance change at different pressures in the low pressure range; (**b**) At different pressures in the high pressure range, a pressure sensor with 12 tex PU yarn correspond to relative current changes; (**c**) Instant response of the CB@PU yarn network, showing a response time of 70 ms.

**Figure 15 sensors-18-04141-f015:**
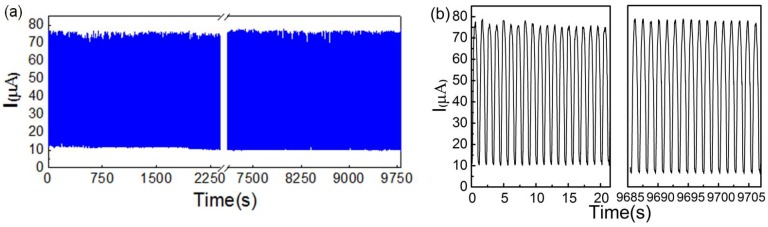
(**a**) Reliability testing of CB@PU yarn network sensors with repeatable loading and unloading pressures of 5 kPa; (**b**) The enlarged view of (**a**) shows a high degree of repeatability and stable sensing capability.

**Table 1 sensors-18-04141-t001:** Resistance value of compressed layer and uncompressed layer under different compression amount.

Compression Amount	Compressed Layer Resistance (KΩ)	Uncompressed Layer Resistance (KΩ)
0	0	130.9
25%	0.028	99.0
50%	0.060	64.3
75%	0.085	32.5
100%	0.110	0

**Table 2 sensors-18-04141-t002:** Fitting parameters of Figure 11.

PU Yarn Linear Density (tex)	Parameters		
	GF (1/cm)	φ_c_	R^2^
12	−0.8842	−4.5025	0.9934
34	−0.9207	−5.7600	0.9921
73	−0.8689	−6.5401	0.9901

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
