# Peer review of "A High Compressibility Pressure—Sensitive Structure Based on CB@PU Yarn Network"

_sensors, 2018, doi:10.3390/s18124141_

Reviewer 1 Report

Dear authors

I have found interesting your proposal of pressure-strain sensor based on polyurethane-Carbon Black particles. However, I have founds several flaws in your manuscript from multiple standpoints: experimental procedures, statements, and results. They are next enumerated:

Flaws related to statements:

·         Lines 31 and 32 read as: “current piezoresistive sensors cannot meet the needs of some emerging applications which require significant elastic compression”.  This statement supports the realization of your work which is OK to mention in the introduction section, but the authors failed to mention some of these emerging applications which require elastic compression. Are these applications related to biomedical or robotic developments? Please cite some work that supports this statement.

·          Lines 61 through 63 read as: “In previous studies, polyurethane pressure sensors were mainly manufactured in forms of sponge or foam, but polyurethane yarn had been rarely reported by far”. If you look up in the internet “polyurethane yarn carbon black”, a vast list of applications reporting this combination is shown. Polyurethane yarn carbon black has been reported in fabrics for conductive heating [1], electromagnetic shielding [2] and also in strain sensing applications [3]. So clearly your statement is unsupported. In order for this article to be accepted for publication, the authors must provide a thorough review of the multiple applications reporting the usage of polyurethane yarn carbon black, i.e. the introduction must be revised.

·         Although not mandatory, it is advisable that at the end of the introduction section the authors provide a scope of the article remainder.

Flaws related to experimental procedure:

·         In Section 2.3, the equipment Fluke 124 is not an oscilloscope. The Fluke 124 is a scopemeter. Oscilloscopes can not be used for resistance measurements but a scopemeter can perform this task. Please provide a connection diagram for the equipment employed during characterization, this may help for a better understanding on how the Fluke 124 and Keithley 2420 were combined to obtain resistance measurements.

·         Figure 7 describes how electrodes are installed on each side of the yarns. This figure is OK, but this figure must be located earlier in the article, because the reader may not understand how experimental data from Figure 5 was obtained. In other words, Figure 7 describes the experimental setup for resistance measurements, but resistance measurements were previously provided in Figure 5. Explanations for Figure 5 need to be reallocated as well.

·         Figure 6 describes the occurrence of quantum tunneling during different stages of yarn compression; this diagram is OK but not supported by experimental results. An easy way to detect quantum tunneling is to perform a Voltage-Current (V-I) test at constant stress/strain. If the V-I curve is linear, then quantum tunneling is not occurring, but if the V-I curve exhibits a quadratic or exponential behavior, then quantum tunneling is occurring [4, 5, 6]. Please provide a method to support the statement from Figure 6.

Flaws regarding results and modelling:

·         What is “Rs” in equation (1); this variable was never introduced.

·         Equation (1) was derived later in Equation (2). This evidence a disordered structure of the article, please correct.

·         In the caption of Figure 8 please indicate what the fitting equation for the plots is.

·         In Figure 10a, the y-axis has no units or dimensions.

·         In Figure 10b, Why the lower plot has only three loading-unloading cycles and the other plots have 6 cycles?

·         According to the mechanical data provided in the paper, could you calculate the mechanical impedance of the pressure sensitive yarn?

Could you estimate the electical hysteresis in the strain-resistance plot? or the electrical resistance in the stress-resistance plot?

In line 255, I guess you meant to say “non-uniformity”, please correct if needed.

References

[1]          Pahalagedara LR, Siriwardane I, Tissera ND, Wijesena RN, de Silva KMN. Carbon black functionalized stretchable conductive fabrics for wearable heating applications. RSC Adv. 2017;7:19174–19180. Available from: http://­dx.doi.org/­10.1039/­C7RA02184D.

[2]          Tian M, Du M, Qu L, Chen S, Zhu S, Han G. Electromagnetic interference shielding cotton fabrics with high electrical conductivity and electrical heating behavior via layer-by-layer self-assembly route. RSC Adv. 2017;7:42641–42652. Available from: http://­dx.doi.org/­10.1039/­C7RA08224J.

[3]          Wang Z, Huang Y, Sun J, Huang Y, Hu H, Jiang R, et al. Polyurethane/Cotton/Carbon Nanotubes Core-Spun Yarn as High Reliability Stretchable Strain Sensor for Human Motion Detection. ACS Applied Materials & Interfaces. 2016;8(37):24837–24843. PMID: 27558025. Available from: https://­doi.org/­10.1021/­acsami.6b08207.

[4]          Celzard A, Furdin G, Mareche JF, McRae E. Non-linear current-voltage characteristics in anisotropic epoxy resin-graphite flake composites. Journal of Materials Science. 1997 Apr;32(7):1849–1853. Available from: https://­doi.org/­10.1023/­A:1018504906935.

[5]          Oskouyi AB, Uttandaraman, Sundararaj, Mertiny P. Current-voltage characteristics of nanoplatelet-based conductive nanocomposites. Nanoscale Res Lett. 2014;9(1):369.

[6]          Paredes-Madrid L, Palacio CA, Matute A, Parra Vargas CA. Underlying Physics of Conductive Polymer Composites and Force Sensing Resistors (FSRs) under Static Loading Conditions. Sensors. 2017;17(9):2108.

Author Response

We have studied your comments carefully and have made revision which marked in blue in the Word. We have tried our best to revise our manuscript according to the comments. Attached please find the revised version, which we would like to submit for your kind consideration.

Reviewer 2 Report

Xingtong Chen and coauthors described highly sensitive pressure sensor based carbon black (CB)-polyurethane (PU) composites in this article. The authors experimentally analyzed the factor determining the pressure sensing capability of the carbon black on PU when deformed. Although the CB@PU composite was developed and analyzed in previous report (Advanced Functional Materials, Wu, X. et al. 26, 6246, 2016), the authors modified the structure to be more sensitive and deformable than before. The reviewer thinks that this article lack the novelty but is acceptable. This article should be published after revisions. The following revisions are needed.

Comment #1: In the line (148), where is the stretching data of PU yarn? When stretched up to 600% strain, how is the pressure sensitivity? Is it stable? Generally, when we stretched it over plastic deformation region, the polymer cannot be perfectly recovered. It means that the sensitivity would be changed and not reliable. What is the maximum stretchability of that? The authors should add the data to this manuscript and address the details to make it clear.

Comment #2: In Fig. 6, the authors suggested the schematic for the percolations. Is it correct? Do the authors have direct evidence? The reviewer is not totally persuaded. As you may know, the previous report proved the percolation in the sponge structure. However, the yarn structure is not the same case. The authors should describe and prove what authors suggested.

Author Response

We have studied your comments carefully and have made revision which marked in blue in the Word document. We have tried our best to revise our manuscript according to the comments. Attached please find the revised version, which we would like to submit for your kind consideration.

Reviewer 3 Report

The manuscript “A High Compressibility Pressure – Sensitive Structure Based on CB@PU Yarn Network” introduced a method to fabricate pressure sensors. The polyurethane yarn was interweaved in a porous ring. Afterward, the whole device was cyclically immersed into carbon black suspensions in cellulose nanowhisker and chitosan solutions. After characterizations, the pressure sensor was demonstrated with wide detection range ( from 25 Pa to 20 kPa), high sensitivity (2.3 kPa-1), short response time (70 ms), and good durability. The topic is interesting and this method may find wide applications to fabricate pressure sensors. However, there are some issues need to be addressed before publication.

1. There are too many grammar errors and inadequate words, which should be fully addressed. For example, the tense of the sentences should be kept consistent with past tenses when discussing the experiments and results.

2. What are the solvents for the CNC and CS solutions? What’s the CS concentration?

3. When the pressures were applied on the device, the electrical resistance of single yarn increased due to the stretching. However, the resistance between yarns may decrease due to the shorting connection. In fact, the resistivity of the yarn increased as the applied pressure increasing, while the effective length decreased when the yarns connected to each other. How these two conflict factors affect the resistance of the whole device? Theoretically, there could be no signal when the effects of the two factors equal. Also, when a tiny pressure applied, the resistance of the device should increase due to the stretching without any contacting. More experiments should be done to explore the effects of process-related parameters such as immersing cycles, and CB concentrations. Besides, the explanation of the pressure-sensing mechanism is too vague.

4. The maximum elongation of the single yarn is given to be 20.6%. How this number comes? If this value is true, most of the data in Fig 5 is useless since the maximum strain is 20.6%.

5. What’s the meaning of the calculations between line 183 and 186? It’s hard to understand. Why the corresponding resistant change rate is not the slope of the plot in Fig.5, which should be (6.75-5.34)/(141.2-100).

6. From Fig.9, the resistance change for the sensor under 20 kPa was about -90%, which means that the current change should be about 9. However, the values in Fig. 10 (b) are about 16. Why is there such a large difference?

7. What’s the meaning of the current signal in Fig. 10 (a)? What’s the applied voltage during the tests?

8. The “standard factor” in line 285 should be gauge factor.

Author Response

We have studied your comments carefully and have made revision which marked in blue in the Word document. We have tried our best to revise our manuscript according to the comments. Attached please find the revised version, which we would like to submit for your kind consideration.

Round  2

Reviewer 1 Report

Dear authors

The overall quality of the manuscript has improved. However, there are still some issues that must be addressed before acceptance.  They are next listed:

It is not advisable that you cite the following reference in the manuscript. It is desirable that you use either: journal papers, conference proceedings or books:

Reference: The research of the extraction of small diameter single-walled carbon nanotubes and large deformation strain sensor [D]. Tianjin University of Technology, 2017.  I tried to look for this article on the web but it was impossible, that's why I recommend not to include it as a reference.

Fig. 3 (a-b) was mentioned on the reviewer's notes, but it was not included on the revised version of the manuscript. I understand that non-linearity was not observed in the V-I test, but be sure to mention this on the final article version. The inclusion of Fig. 3 is not mandatory because it shows a perfect linear V-I response, but it is required to state that a V-I test was performed (and include the results in a written-basis)

For better comprehension, experimental results from Figure 14 can be split into multiple figures. Fig 14a is represented in percentage notation, but Figure 14b is not. Try to standardize the axes notation along the figures.

Author Response

We thanks for your constructive criticisms that have helped us to improve our manuscript. We have studied your comments carefully and have made revision which marked in blue in the Word file. We have adjusted the figures and added the necessary instructions and content in the paper. Attached please find the revised version, which we would like to submit for your kind consideration.

At last, special thanks to you for your good comments!

Reviewer 3 Report

Since the authors have addressed all the comments, I support the publication of this manuscript at the current version.

Author Response

Thank you for your guidance on this article. With the help of you, the quality of this article has been greatly improved. We would like to express our great appreciation to you for comments on our paper. Once again, thank you very much for your comments and suggestions!